Impact of Argemone mexicana L. on tomato plants infected with Phytophthora infestans

Hernández-Soto Iridiam 1
González-García Yolanda 2
Juárez-Maldonado Antonio 3
Hernández-Fuentes Alma Delia almah@uaeh.edu.mx 1
1 Universidad Autónoma del Estado de Hidalgo, Instituto de Ciencias Agropecuarias , Tulancingo de Bravo , Hidalgo , Mexico
2 Centro de Investigación Regional Noreste, Campo Experimental Todos Santos, Instituto Nacional de Investigaciones Forestales, Agrícolas y Pecuarias , Emiliano Zapata , La Paz, B.C.S , Mexico
3 Universidad Autónoma Agraria Antonio Narro, Departamento de Botánica , Saltillo , Coahuila , Mexico
Adhikari Tika
Electronic publication date: 2024 Jan 3
Publication date: 2024
Volume: 12
Electronic Location ID: e16666
Received 2023 Jun 29; Accepted 2023 Nov 21
Copyright: ©2024 Hernández-Soto et al.
Copyright year: 2024
Copyright holder: Hernández Soto-et al.
License: This is an open access article distributed under the terms of the Creative Commons Attribution License, which permits unrestricted use, distribution, reproduction and adaptation in any medium and for any purpose provided that it is properly attributed. For attribution, the original author(s), title, publication source (PeerJ) and either DOI or URL of the article must be cited.
License URL: https://creativecommons.org/licenses/by/4.0/

Keywords: Antioxidants, Biofungicide, Biostimulant, Defense compounds, Biotic stress

Funding: Consejo Nacional de Humanidades, Ciencias y Tecnologías (Conhacyt) 715170 This work was supported by the Consejo Nacional de Humanidades, Ciencias y Tecnologías (Conhacyt) by a scholarship (715170) awarded to Iridiam Hernández-Soto. The funders had no role in study design, data collection and analysis, decision to publish, or preparation of the manuscript.

==============================
Background

Fungal diseases can cause significant losses in the tomato crop. Phytophthora infestans causes the late blight disease, which considerably affects tomato production worldwide. Weed-based plant extracts are a promising ecological alternative for disease control.

Methods

In this study, we analyzed the plant extract of Argemone mexicana L. using chromatography-mass spectrometry analysis (GC-MS). We evaluated its impact on the severity of P. infestans, as well as its effect on the components of the antioxidant defense system in tomato plants.

Results

The extract from A. mexicana contains twelve compounds most have antifungal and biostimulant properties. The findings of the study indicate that applying the A. mexicana extract can reduce the severity of P. infestans, increase tomato fruit yield, enhance the levels of photosynthetic pigments, ascorbic acid, phenols, and flavonoids, as well as decrease the biosynthesis of H2O2, malondialdehyde (MDA), and superoxide anion in the leaves of plants infected with this pathogen. These results suggest that using the extract from A. mexicana could be a viable solution to control the disease caused by P. infestans in tomato crop.

Introduction

Tomato (Solanum lycopersicum L.) is one of the most important crops produced and consumed worldwide (Nkongho et al., 2022) due to its high content of vitamin A and C, phosphorus, iron, beta-carotene, anthocyanins, and lycopene (Saffan et al., 2022). According to the latest data from FAOSTAT, 177 million tons of tomatoes are produced annually worldwide (Nkongho et al., 2022). However, crops diseases caused by phytopathogenic oomycetes cause losses of between 20% and 40% of total tomato production (Ali et al., 2020). One of the diseases that affects tomato production is caused by Phytophthora infestans.

P. infestans is a heterothallic, hemibiotrophic oomycete responsible for late blight disease (Kumar et al., 2022; Dong & Zhou, 2022). The infection spreads by air to other plant tissues, reaching total necrosis of infected plants within five to 10 days (Nowicki et al., 2012). Control of late blight can sometimes be achieved through a single procedure, but in most cases, it requires the use of integrated pest and disease management (IPM; (Ramasamy & Ravishankar, 2018) to minimize the use of pesticides (Zhang, Islam & Liu, 2022). Although chemical fungicides remain the most effective strategy to control this disease, excessive use of chemical fungicides can generate health problems, pathogen resistance, and environmental contamination (Bouket et al., 2022), leading to a growing need to explore and generate organic products to control plant pathogens.

Weed-based fungicides are gaining popularity in organic agriculture because they are safe to use on crops grown for human consumption. There is currently a lucrative market of consumers willing to pay more for organically produced foods (Ngegba et al., 2022). There is an excellent opportunity to create more biofungicides using plant extracts (Borges et al., 2018). Botanical extracts are currently being used in organic agriculture to increase yield and quality in crop production, primarily in Europe, though researchers worldwide are promoting their use in crop production (Osman, Mohamed & Farag, 2021).

A weed plant is any species that competes for space, nutrients, and solar energy with a crop of commercial interest (Horvath et al., 2023). One common weed plant is Argemone mexicana L., commonly known as chicalote. This species contains berberine, dehydrocorydalmine, allocryptopin, oxyberberine, cysteine, and phenylalanine, to which different biological activities are attributed (Brahmachari, Gorai & Roy, 2013).

Some reports of weed plant extracts being used as biofungicides and/or biostimulants. Hanaa et al. (2011) used extracts of neem (Azardiachta indica A. Juss) in tomato seedlings infected with Fusarium oxysporum. They reported a significant increase in the growth of shoots and roots of tomato seedlings, in addition to an inhibition in the growth of this phytopathogen. Tighe Neira et al. (2013) evaluated the effect of nettle extract (Urtica dioica L.) and thorny broom (Ulex europaeus L.) on chili seedlings (Capsicum annuum L.). Both extracts generated affected the biomass production and concentration of phenolic compounds. Jasso de Rodríguez et al. (2020) reported that sumac (Rhus muelleri F. A. Barkley) extract increased stem length and diameter, the dry weight of leaves, the number and weight of fruits, and fruit production in tomato plants.

The present study aimed to evaluate the antifungal activity of Argemone mexicana L. extract using chromatography-mass spectrometry analysis (GC-MS) and research its role in improving the physiological and biochemical effects of tomato plants infected with Phytophthora infestans.

Materials & Methods

Plant material

Random samples were collected of Argemone mexicana stems with leaves in the vegetative development stage during the winter period in the Cuautepec region of Hinojosa, Hidalgo, Mexico. The sample area was located at 20°09′00″N, 98′00″W., at an altitude between 2,200–2,900 m above sea level with an average annual temperature of 20 °C (Fig. 1). The collected samples were collected into plastic bags and transported to the postharvest laboratory of the Universidad Autonoma del Estado de Hidalgo (UAEH). Immediately upon arrival, the leaves were separated from the stems and stored at −70 °C (Thermo Scientific 703 Ultra-Low Freezer, Grand Island, NY, USA) and then preserved in a freeze dryer (Model 79480; Labconco Corporation, Kansas City, MO, USA). The leaves were subsequently ground in a blade mill (GM 200, Grindomix, Glen Mills Inc, Clifton, NJ, USA) and the samples were stored at 5 °C until further analysis. A complete plant was conserved for identifying the species, which was carried out in the botany laboratory of the Institute of Biological Sciences of the Universidad Autonoma del Estado de Hidalgo.

Figure 1 Argemone mexicana L. plant material collection site.

Map data ©2023 Google.

Obtaining the plant extract

The leaf extract was obtained by maceration with ethyl acetate (200 mL) and 20 g of plant sample (1:10). The maceration was maintained for seven days, after which the extract was filtered twice through Whatman no. 1. The solvent in the extract was removed in a vacuum, using a rotary evaporator (Büchi R-215; Buchi AG, Flawil, Switzerland) for four hours at a temperature of 40 °C and a pressure of 60 mbar, as indicated by the instrument. The extract was stored in a desiccator at 26 °C and 0% relative humidity (RH) until its use in bioassays, following the methodology described by Jasso de Rodríguez et al. (2011). This procedure was repeated throughout the experiment to ensure an adequate supply of the plant extract for the different research applications. On average, extract was 60% (12 g of solid material). Before the bioassays, in vitro microbiological tests were carried out following the methodology of Vásquez Covarrubias et al. (2013) over seven days, where different concentrations were tested: 1,000, 1,500, 2,000, 2,500, 3,000 and 3,500 mL−1. The concentration of 3,500 mg L−1 showed fungicidal activity against P. infestans, so this concentration was used for the greenhouse bioassays.

Chromatography-mass spectrometry analysis (GC-MS)

The chromatographic analysis was conducted on an Agilent Technologies 7890A GC system connected to a 5975 GC/MSD in split-less scan mode. The separation of metabolites in the extracts was performed on a DB17HT column (30 m × 0.25 mm 1D × 0.15 µL) and an ionization system at an energy of 70 eV. Helium was the carrier gas with a constant flow of 3 mL/min. An injection volume of 1 µL was used at an injection temperature of 270 °C. The oven heating program was isothermal: one minute at 60 °C, followed by one minute at 30 °C, one minute at 80, one minute at 10 °C, one minute at 110 °C, one minute at 6 °C, and finally 270 °C sustained for 40 min. The mass spectra of the compounds were interpreted using the National Institute of Standards and Technology (NIST) database, following the methodology proposed by Raja Rajeswari, RamaLakshmi & Muthuchelian (2011). The results reported were: retention times (RT) in minutes, name, molecular formula, molecular weight, and area (%) of the components. The analyzed solutions were prepared by taking 1 g of each dried extract and solubilizing it in 20 mL of ethyl acetate solvent. The solutions were filtered through Whatman filter paper No. 1 to remove any particle solids and ensure that the solutions used were clear and transparent. All chemicals used were high purity, and analytical grade with according to the methodology of Ullah et al. (2018).

Crop development and management

The culture was established in a greenhouse with a polyethylene cover. Saladette ‘El Cid F1’ tomato seeds (Harris Moran, Davis, CA, USA) with indeterminate growth were transplanted into 12-L black polyethylene bags using a mixture of peat and perlite as substrate in a 1:1 ratio (v/v). The tomato plant was grown on a single stem. For crop nutrition, an irrigation system was used at different concentrations during different growth periods: 25% in the vegetative stage, 50% in the flowering stage, 75% during fruit setting, and 100% during fruit filling and harvesting, according to the methodology described by Steiner (1961). The temperature and humidity of the greenhouse were constantly monitored with the help of a digital hygrometer. Temperatures ranged between 22 ± 2 °C and humidity was maintained at ≥70%. Greenhouse windows were opened or closed, and irrigation was performed in hallways to maintain these temperature and humidity conditions. A drip irrigation drainage system was used with different levels of irrigation at each growing phase taking into account the dynamics in the management, environment, and development of the crop: 100 mL for the germination stage, 500 mL for the vegetative development stage, 1,500 mL for the for flowering and fruit set stage, and 2,500 mL for the fruiting stage, according to the methods outlined by Flores (2007).

Preparation of the inoculation and evaluation of the severity of the disease

The inoculation was prepared following the method described by Smith, Hammerschmidt & Fulbright (1991), with some modifications. P. infestans was obtained from “Centro de Ciencias Agrarias de la Universidad Autonoma de Aguascalientes, Mexico”. This was propagated on potato dextrose agar (PDA) and incubated for 18 days at 27 °C. The fungal growth, together with the PDA and sterile distilled water, was mixed, placed in a flask, and shaken. Then the mixture was filtered through sterile gauze and the mycelium was collected. The liquid from the petri dishes was concentrated and a spore count was performed in the Neubauer chamber to adjust to a concentration of (1 ×106 sporangia mL−1). Tomato plants with young and developed second and third leaves were inoculated with the sporangia suspension (two mL per plant) 30 days after transplanting using a camel-hair brush. The plants were covered with a perforated plastic bag to achieve a relative humidity of ≥70% around the foliage, as proposed by the methodology of Ortiz et al. (2016). The severity scale of P. infestans was determined using the method described by Zárate-Martínez et al. (2018), with some modifications adapted to this pathogen. This scale has values from 0-5: 0: leaves show no signs of wilting; 1: 1–10% of the leaves show slight marginal wilting; 2: 11–25% of the leaves are wilted; 3: 26–49% of the leaves show wilting and fruit lesions; 4: 50–74% of the leaves show wilting with partial leaf fall; and 5: all leaves are withered. The severity index was calculated using the formula described by Raupach et al. (1996):

IS = [∑ (NC ×NPC) ×100%]/ (NTP ×CMA)

where: IS = severity index; NC = classification number; NPC = number of plants in the classification; NTP = total number of plants; CMA = highest classification.

Application of treatments

Five treatments were considered: plants without inoculation and chicalote extract application (T1), inoculated plants and extract application (T2), plants inoculated with the pathogen and commercial fungicide captan (T3), plants inoculated with the pathogen without additional treatment, (T4) and plants without inoculation or any additional treatment (T5). Each tomato plant in each application was sprayed using a multipurpose manual spray pump with an extract solution at a concentration of 3,500 mg L−1 solubilized in 100 mL of water and combined with 1.5 mL of Bionex® adherent. After one week of inoculation, 100 mL−1 of the commercial fungicide captan was applied to each plant, followed by four more applications, each two weeks apart, based on the dosage recommended by the product (3 × 106 mg L−1) and following the methodology described by El-Nagar et al. (2020) and Jasso de Rodríguez et al. (2020) (Fig. 2).

Figure 2 Application of treatments, data collection, and sampling in the different phenological stages of the tomato crop.

Made with ©Canva 2023.

Agronomic analysis

To evaluate the effect of the treatments on the agronomic variables of the tomato plants (average weight of the fruits, number of fruits per plant, average weight of the fruit per plant, stem diameter, dry weight of the aerial, and root biomass), measurements were taken one week after each of the five applications (Fig. 2). When the plants had six clusters, the growth apex of the plants was removed to facilitate crop management. Stem diameter was measured with a digital vernier caliper between the first and second leaves at the base of the plant 105 days after transplanting. Fruit yield, average weight of the fruits, and the number of fruits harvested were calculated based on the data of the five samplings during the experimentation time. The plants were cut on the surface of the substrate 105 days after transplanting. The dry weight of the roots and shoots (stems and leaves) was measured after they were dried in a drying oven (Model HFA-1000DP; CRAFT, CDMX, Mexico) for 72 h at a constant temperature of 80 °C, according to the methodology described by Hernández-Hernández et al. (2018).

Sampling of leaves and fruits

Sampling was carried out one week after each of the five applications (Fig. 2); the samples comprised three plants per treatment for each block and four fully expanded young leaves from each plant (2nd and 3rd leaves). Starting 60 days after transplanting, fruit harvest was carried out each week, with fruits harvested when they had a commercial maturity index (completely colored) at ripening stage six according to the scale of the United States Department of Agriculture (USDA, 2021). Samples were stored at −70 °C (Thermo Scientific 703 Ultra-Low Freezer; Thermo Fisher Scientific, Waltham, MA, USA) and subsequently lyophilized (Freeze Dryer, model 79480; Labconco Corporation, Kansas City, MO, USA) and macerated until a fine powder was obtained. This sample was then used to determine photosynthetic pigments, stress biomarkers, and non-enzymatic antioxidant compounds.

Photosynthetic pigment measurements

The concentrations of chlorophyll a and b, and total chlorophyll were analyzed in lyophilized leaves. A mix of 10 mg of lyophilized leaves and two mL of hexane: acetone (3:2) was centrifuged at 12,000 rpm for 10 min at 4 °C. The resulting extract was read in a spectrophotometer at 645 and 663 nm wavelengths. The resulting absorbances were used for subsequent calculations with equations proposed by Nagata & Yamashita (1992).

Stress biomarkers test

Hydrogen peroxide (H2O2) was assessed according to the methodology described by Sergiev, Alexieva & Karanov (1997), with some modifications, 10 mg of lyophilized sample was taken and homogenized with 1,000 µL of cold 0.1% trichloroacetic acid in an ice bath. After centrifuging the homogenate at 12,000 rpm for 15 min at 4 °C, 250 µL of the supernatant was mixed with 750 µL of 10 mM potassium phosphate buffer (pH 7.0) and 1,000 µL of potassium iodide (1 M). The absorbance of the supernatant was measured at 390 nm.

The thiobarbituric acid (TBA) test, which determines malondialdehyde (MDA) content as an end product of lipid peroxidation, was used to measure lipid peroxidation in the leaves; MDA was determined according to the methodology described by Heath & Packer (1968), with some modifications. In total, 50 mg of the sample was mixed with 1,000 µL of thiobarbituric acid (TBA) (0.1%) and centrifuged (10,000 rpm, 20 min, 4 °C). Then 500 µL of the supernatant was added to 1,000 µL of TBA (0.5%) in trichloroacetic acid (20%). The mixture was incubated in water at 90 °C for 30 min, the reaction was quenched with ice, and the sample was centrifuged (10,000 rpm, 5 min, 4 °C). Then, the absorbance of the supernatant was measured at 532 nm to calculate the amount of MDA-TBA complex using an extinction coefficient of 155 mM−1 cm−1.

The superoxide anion (O2•−) content was determined according to the methodology described by Wang & Luo (1990), with some modifications. A total of 20 mg of lyophilized sample was added with 5 mg of PVP and homogenized with 1,000 µL of cold 50 mM phosphate buffer (pH 7.8). The mixture was kept incubated for 30 min at 25 °C. Subsequently, 650 µL of the incubated solution was combined with 650 µL of aminobenzenesulfonic acid (17 mM) and 650 µL of 1-naphthylamine (7 mM). The absorbance was read at 530 nm.

Non-enzymatic antioxidant compounds measurement

The content of total phenols was obtained using the methodology outlined by Singleton & Rossi (1965), with some modifications. A total of 100 mg of lyophilized sample and one mL of a water/acetone solution (1:1) was homogenized for 30 s. The sample tubes were centrifuged (17,500 rpm, 10 min, 4 °C), then 18 µL of the supernatant, 70 µL of the Folin–Ciocalteu reagent, and 175 µL of 20% sodium carbonate (Na2CO3) were placed in a test tube, and 1,750 µL of distilled water was added. The samples were placed in a water bath for 30 min at 45 °C. Finally, the reading was taken at a wavelength of 750 nm.

The concentration of total flavonoids was determined by mixing 20 mg of lyophilized tissue with more than two mL of methanol and subsequently filtering with a Whatman Filter No. 1. For the quantification, a mix of one mL of solution and one mL of AlCl3 (2%) was incubated in dark conditions for 20 min. The sample was then read at 415 nm, according to the methodology described by Arvouet-Grand et al. (1994).

Vitamin C content was determined according to the methodology described by Klein & Perry (1982), which was read at 515 nm, and the results were expressed as milligrams per 100 g of dry weight (mg 100 g−1 of DW).

The β-carotene content was determined according to the methods outlined by Zscheile, Comar & Mackinney (1942) and Nagata & Yamashita (1992). The absorbances were read at 453, 505, 645, and 663 nm, and the results were expressed in mg 100 g−1 of DW.

Yellow carotenoids (β-carotene, β-cryptoxanthin, zeaxanthin) were evaluated according to the methods reported by Hornero-Méndez & Mínguez-Mosquera (2001). Yellow carotenoid measurements were expressed as milligrams per 100 g dry weight (mg 100 g−1 DW).

The quantification of proteins was determined using Bradford’s colorimetric technique, a method reported by Bradford (1976). The samples were read at a wavelength of 630 nm on a microplate reader. The total proteins were expressed in mg g−1 of DW.

Statistical analysis

Five replicates with three plants per block and fifteen plants per treatment unit were considered for each of the treatments, in a randomized complete block design. An analysis of variance and Fisher’s least significant difference (LSD) test of means (α = 0.05) were performed to analyze the agronomic and biochemical variables of tomato. To determine the differences between treatments in the severity of P. infestans, a repeated measures multivariate analysis of variance and Hotteling test (α = 0.05) were performed. All statistical procedures were performed using the Infostat 2020 software.

Results

Chromatography-mass spectrometry analysis (GC-MS)

There were twelve compounds identified in this extract, most antifungal and biostimulant activity, including: hexadecanoic acid, methyl ester-; 9,12,15-Octadecatrienoic acid, methyl ester, (Z,Z,Z)-; 2-Propenoic acid, 2-methyl-, 1,2-ethanediyl ester; dl- α-Tocopherol succinate and, 3,6-Dimethyl-4H-furo[3,2-c] pyran-4-one (Table 1).

P. infestans severity and crop development

Disease severity decreased with the application of A. mexicana extract and captan. Late blight reached a severity of 90% in the Infes treatment (Fig. 3A), but the Infes+EXAm treatment decreased disease severity by 48%. In comparison the captan reduced disease severity by 69% during the entire vegetative cycle of the crop. In the first weeks after inoculation, when P. infestans was most effective, the Infes+ EXAm and Infes+Captan treatments reduced disease severity by 57.6% and 71%, respectively, compared to the Infes treatment, During the fruiting stage (75 days after transplanting) the signs of disease were less severe (Fig. 3B). The EXAm and control treatments remained disease-free throughout the evaluation, which was expected since the plants in these treatments were not inoculated with P. infestans (Fig. 3B).

The analysis of agronomic parameters (Fig. 4) showed that the EXAm treatment increased crop yield by 128% compared to the control treatment. The Infes treatment reduced crop yield by 12% compared to the EXAm treatment, but the Infes+EXAm treatment increased crop yield by 127% and the Infes+Captan treatment increased crop yield by 113% compared to the Infes treatment (Fig. 4A). The number of fruits per plant was reduced by 23% with the Infes treatment, by 8% with the Infes+EXAm treatment, and by 16% with the Infes+Captan treatment compared to the EXAm treatment. The number of fruits increased by 115% with the EXAm treatment compared to the control treatment (Fig. 4B). Similar patterns were observed for the average weight of fruits per plant (Fig. 4C). The Infes treatment reduced the average weight of the fruits by 11% compared to the EXAm treatment, but increased fruit weight by 110% compared to the control treatment. Even in diseased plants with the application of extract (Infes+EXAm) and commercial fungicide (Infes+Captan), fruit weight increased an average of 102% in the same direction of comparison.

Table 1 Chemical compounds identified in A. mexicana L. extract.

RT
(min)	Name of the compound	Molecular
formula	MW	Area (%)	
6.22	2-Propyn-1-ol, acetate	C5H6O2	98	1.27	
16.71	Hexadecanoic acid, methyl ester	C17H34O2	270	14.50	
17.76	n-Hexadecanoic acid	C16H32O2	256	13.54	
20.30	9,12,15-Octadecatrienoic acid, methyl ester, (Z,Z,Z)-	C19H32O2	292	23.06	
33.31	2-Propenoic acid, 2-methyl-, 1,2-ethanediyl ester	C10H14O4	198	18.83	
35.87	3,6-Diiodoacridine	C13H7I2N	431	2.75	
35.89	dl- α-Tocopherol succinate	C33H54O5	530	10.48	
40.68	trans-1,2-Cyclohexanedicarbonitrile	C8H10N2	134	2.58	
41.93	Cyclobutane, methyl-	C5H10	70	0.07	
43.57	Benzenemethanamine, N-(1-methylethyl)-	C10H15N	149	0.64	
43.79	3,6-Dimethyl-4H-furo[3,2-c]pyran-4-one	C9H8O3	164	12.05	
44.01	N-(9H-Fluoren-2-ylmethylene)-p-toluidine	C21H17N	283	0.23	
Notes.

RT Retention time

MW Molecular weight

These compounds are the result of 12 replicates per sample.

Figure 3 (A) Severity of P. infestans in tomato plants over 105 days after transplanting. (B) Signs of the disease in plants 75 days after transplanting treated with different treatments at the end of the evaluation.

Infes: plants inoculated with P. infestans; Infes+Captan: inoculated plants with commercial fungicide application; Infes+EXAm: inoculated plants with A. mexicana extract application; EXAm: healthy plants with A. mexicana extract application; Control: healthy plants without any additional application. The bars represent the standard error of the mean. Significance of the analysis of variance of repeated measures (α = 0.05). Different letters between treatments indicate significant differences according to the Hotteling test (α = 0.05). Made with ©Canva 2023.

Figure 4 (A) Fruit yield per plant; (B) the number of fruits per plant; (C) average fruit weight; (D) stem diameter; (E) aerial dry weight; (F) root dry weight.

EXAm: healthy plants with A. mexicana extract application; Infes+EXAm: inoculated plants with A. mexicana extract application; Infes+Captan: inoculated plants with commercial fungicide application; Infes: plants inoculated with P. infestans; Control: healthy plants without any additional application. Different letters in the bars indicate significant differences according to Fisher’s least significant differences test (α = 0.05); n = 5 standard error.

Throughout the vegetative cycle, stem diameter was reduced by an average of 15% in the Infes+EXAm and Infes+Captan treatments. However, stem diameter increased by 110% with the EXAm treatment compared to the control treatment and was reduced by 23% in the Infes treatment compared to the EXAm treatment (Fig. 4D). Although the foliar application of chicalote extract affected crop yield and development, there were no significant differences in aerial (Fig. 4E) and root (Fig. 4F) dry weight.

Photosynthetic pigment content in leaves

The content of photosynthetic pigments was variable throughout the different evaluations in the vegetative cycle of the crop (Fig. 5). Chlorophyll a content (Fig. 5A) increased by 157% with Infes+EXAm compared to the control at 45 ddt (flowering stage and fruit set). At 60 and 75 ddt (flowering stage and fruit set, fruiting stage) EXAm and Infes+EXAm increased chlorophyll a content by 129% and 121%, respectively, compared to the control. Still, chlorophyll a was reduced by 35% with the Infes+Captan treatment in the same time comparison. On days 90 and 105 after transplanting (fruiting stage), the chlorophyll content was reduced by 52% with the Infes treatment, but increased by 150% with the EXAm treatment compared to the control treatment.

Figure 5 Chlorophyll content in tomato leaves with different treatments.

(A) Chlorophyll a; (B) chlorophyll b; (C) total chlorophyll. EXAm: healthy plants with A. mexicana extract application; Infes+EXAm: inoculated plants with A. mexicana extract application; Infes+Captan: inoculated plants with commercial fungicide application; Infes: plants inoculated with P. infestans; Control: healthy plants without any additional application. Different letters in the bars indicate significant differences according to Fisher’s least significant differences test (α = 0.05); n = 5 standard error.

Chlorophyll b content (Fig. 5B) decreased by an average of 47% with the Infes+Captan treatment compared to the control treatment throughout the different evaluations of the vegetative cycle. Between 60 and 75 ddt (flowering stage and fruit set, fruiting stage), chlorophyll b content was reduced by 78% with the Infes+Captan treatment. In these same evaluations, the EXAm and Infes+EXAm treatments increased chlorophyll b content by 117% and 133%, respectively, compared to the control treatment. Between 90 and 105 ddt (fruiting stage), chlorophyll b content was reduced by 48% with the Infes treatment, but increased by 138% with the EXAm treatment compared to the control treatment.

In general, the highest values of total chlorophyll content (Fig. 5C) were seen in the control treatment except for 45 ddt (flowering and fruit set stage) and at 60, 75, and 90 after transplantation (flowering and fruit set stage, fructification stage) when the EXAm and Infes+EXAm treatments increased total chlorophyll content by 129%, 154%, 110%, and 124%, respectively, compared to the control treatment. Throughout the different evaluations during the vegetative cycle, the total chlorophyll content decreased considerably with the Infes+Captan treatment. At 60 and 75 ddt (flowering and fruit set, fructification stage), total chlorophyll content was reduced by 68% with the Infes+Captan treatment compared to the control treatment. Between days 90 and 105 after transplanting (fruiting), total chlorophyll content was reduced with the Infes+EXAm, Infes+Captan, and Infes treatments by 48%, 47%, and 52%, respectively, compared to the control treatment, and increased by 148% with the EXAm treatment compared to the Infes treatment.

Stress biomarkers in tomato leaves

The hydrogen peroxide (H2O2) content in tomato leaves showed significant differences between treatments in some evaluation periods (Fig. 6A). Although there were no significant differences in H2O2 content between treatments at 45 ddt (flowering stage and fruit set), between 60 and 75 ddt (flowering stage and fruit set, fructification stage), the H2O2 content increased with the Infes+EXAm and Infes treatments by 151% and 200%, respectively, compared to the EXAm treatment, which was the treatment with the lowest H2O2 content with 10% less than the control treatment. On days 90 and 105 ddt (fruiting) the highest values of H2O2 occurred with the Infes treatment, but H2O2 content was reduced by 58% with the EXAm treatment, by 60% with the Infes+EXAm treatment, and by 65% with the control treatment compared to the Infes treatment.

Figure 6 (A) Hydrogen peroxide (H2O2); (B) malondialdehyde (MDA); and (C) superoxide anion in tomato leaves with different treatments.

EXAm: healthy plants with A. mexicana extract application; Infes+EXAm: inoculated plants with A. mexicana extract application; Infes+Captan: inoculated plants with commercial fungicide application; Infes: plants inoculated with P. infestans; Control: healthy plants without any additional application. Different letters in the bars indicate significant differences according to Fisher’s least significant differences test (α = 0.05); n = 5 standard error.

Malondialdehyde (MDA) content (Fig. 6B) did not differ significantly between treatments, Still, at 45 ddt (phenological flowering and fruit set), MDA content was reduced by 35% and 28% with the EXAm and Infes+EXAm treatments, respectively, compared to the control treatment. Between 60 and 75 days after transplanting (phenological flowering and fruit set, fructification stage), as well as between 90 and 105 ddt (fruiting), MDA synthesis increased with all treatments previously inoculated with P. infestans. In contrast EXAm and the control treatment reported the lowest concentrations of MDA, as expected. The Infes+EXAm, Infes+Captan, and Infes treatments increased MDA content by 133%, 120% and 145%, respectively, compared to the control treatment between 60 and 75 ddt (flowering stage and fruit set, fructification stage). Between 90 and 105 ddt (fruiting) the Infes+Captan treatment increased the MDA content even more—by 17% and 18%–compared to the Infes+EXAm and Infes treatments, respectively.

In most evaluations during the vegetative cycle of this crop, the superoxide anion concentration (Fig. 6C) increased with the Infes treatment, increasing an average of 145% at 45 ddt (flowering stage and fruit set) and between 60 and 75 ddt (flowering stage and fruit set, fructification stage) compared to the control treatment. At 45 ddt (flowering stage and fruit set), the superoxide anion content was reduced by 26% and 6% with the EXAm and Infes+EXAm treatments, respectively, compared to the control and by 41% and 28%, respectively, compared to the Infes treatment. Between 60 and 75 ddt (flowering and fruit set stage, fruiting) and between 90 and 105 ddt (fruiting), the EXAm and/or Infes+EXAm treatments reduced the concentration of superoxide anion by an average of 5% and 8%, respectively, compared to the control treatment. Between 90 and 105 ddt (fruiting stage), the control treatment increased superoxide synthesis by 114% compared to the EXAm treatment.

Non-enzymatic antioxidant compounds

The phenol content was reduced by an average of 21% with the Infes treatment compared to the control treatment throughout all evaluations during the vegetative cycle (Fig. 7A). In some evaluations, the difference was considerable. For example, at 45 ddt (flowering and fruit set stage) phenol content was reduced by 43% with the Infes treatment compared to the control treatment. During the flowering and fruit setting stage (60 days after transplanting), the content of phenolic compounds increased by 127% with the EXAm treatment compared to the control treatment. Flavonoid content differed significantly between the treatments (Fig. 7B), with the Infes+Captan and Infes treatments reducing flavonoid content by 22% and 25%, respectively, compared to the control treatment, and the EXAm treatment increasing flavonoid content by 113% on average compared to the control treatment at the different evaluation times. Vitamin C content did not change with the presence of P. infestans (Fig. 7C). The β-carotene content increased by 106% with the Infes+EXAm treatment compared to the control treatment 60 and 75 days after transplanting (phenological stage of flowering and fruit set, fruiting; Fig. 7D). The Infes+Captan treatment reduced β-carotene content an average of 38% compared to the control treatment. Between 60 and 75 days after transplanting (phenological stage of flowering and fruit set, fruiting), yellow carotenoids (Fig. 7E) were reduced by an average of 30% with the Infes+Captan treatment compared to the control treatment, and increased by 113% with the EXAm treatment compared to the control treatment. The protein content (Fig. 7F) increased with the EXAm and Infes+EXAm treatments, by 127% and 124%, respectively, compared to the control treatment at 60 and 75 ddt (phenological stage of flowering and fruit set, fruiting stage). At 90 and 105 ddt (fruiting stage), the Infes treatment reduced the protein content by 30% compared to the control treatment.

Figure 7 Antioxidant compounds in tomato leaves with different treatments.

(A) Phenols; (B) flavonoids; (C) vitamin C; (D) β-carotenoids; (E) yellow carotenoids; (F) proteins. EXAm: healthy plants with A. mexicana extract application; Infes+EXAm: inoculated plants with A. mexicana extract application; Infes+Captan: inoculated plants with commercial fungicide application; Infes: plants inoculated with P. infestans; Control: healthy plants without any additional application. Different letters in the bars indicate significant differences according to Fisher’s least significant differences test (α = 0.05); n = 5 standard error.

Discussion

Chromatography-mass spectrometry analysis (GC-MS)

In the Argemone mexicana L. extract, the following compounds were found, some with previous reports of biological and biostimulant activity (Table 1). Still, it is important to mention that these compounds have not been tested against P. infestans; and although 9,12,15-Octadecatrienoic acid, methyl ester, (Z,Z,Z) has been evaluated as biostimulant, tomato crop has not been considered in the evaluation; which makes this research the first report of A. mexicana extract as a potential antifungal against late blight and as a biostimulant in tomato plants. According to the literature review, hexadecanoic acid, methyl ester has been reported to have antifungal activity against phytopathogenic fungi such as Alternaria solani (Abubacker & Deepalakshmi, 2013) and has been reported to be involved in plant signaling pathways in response to stress, and in stimulating growth in Perilla sp (Ahn et al., 2020); n-hexadecanoic acid has been reported to have antibacterial activity and DPPH and H2O2 radical scavenging activity (Saravanakumar et al., 2018); 9,12,15-Octadecatrienoic acid, methyl ester, (Z,Z,Z) present in botanical extracts of Linum usitatissimum L. has been reported to have potential antifungal and biostimulant activity in soybean crop yield (Kocira et al., 2021); 2-Propenoic acid, 2-methyl-, 1,2-ethanediyl ester has shown antifungal activity against phytopathogens such as Rhizoctonia solani, Sclerotium rolfsii, and Fusarium oxysporum (Septiyanti et al., 2019); dl- α-Tocopherol succinate, one of the most potent antioxidants, prevents the spread of lipid peroxidation (Munné-Bosch, 2005); and 3,6-Dimethyl-4H-furo[3,2-c]pyran-4-one has been shown to attract pollinating wasps (Freitas et al., 2020).

P. infestans severity and crop development

A 90% of the disease severity in tomato plants inoculated with Phytophthora infestans was related to effector proteins that manipulate the physiology of the host plant to facilitate pathogen colonization (Whisson et al., 2007; Haas et al., 2009). The plant immune response can trigger programmed cell death to limit pathogen’s spread (Jones & Dangl, 2006). Still, in this experiment, programmed cell death increased the infectivity of P. infestans because P. infestans has a biotrophic and a necrotrophic phase during its life cycle (Fig. 3) (Zuluaga et al., 2016).

The foliar application of A. mexicana extract on inoculated plants reduced disease severity by 48% (Fig. 3A), likely due to compounds present in the extract, such as: Hexadecanoic acid, methyl ester; 9,12,15-Octadecatrienoic acid, methyl ester, (Z,Z,Z); 2-Propenoic acid, 2-methyl-, 1,2-ethanediyl ester, which have all been reported to have antifungal activity (Abubacker & Deepalakshmi, 2013; Kocira et al., 2021; Septiyanti et al., 2019). Constituents of plant origin can exhibit different modes of action against phytopathogens, including: interference with phospholipid cell membranes, which results in increased permeability profile and loss of cellular constituents (Omojate Godstime et al., 2014); inhibition of cellulase synthesis, chelation of metals necessary for the activity of microbial enzymes, and polymerization into crystalline structures that can act as a physical barrier during pathogen attack (Skadhauge, Thomsen & Von Wettstein, 1997); disruption of the electron transport chain and the slowing of all ATP-dependent functions, inhibition of DNA synthesis and helicase activity, compromising cell division, and termination of chromosome replication resulting in inhibition of the growth in phytopathogens (Fontana et al., 2022).

The presence of P. infestans affects crop development and yield, as well as the quality of the fruit, with total yield losses in tomatoes reaching up to $112 million annually (Nowicki et al., 2012). It is estimated that the costs of late blight control in tomato exceed $5 billion annually worldwide (Galeano Garcia et al., 2018). The application of A. mexicana extract increased the yield of tomato plants by 146% compared to the yield of plants inoculated with P. infestans (Fig. 4A). Turóczi et al. (2020) evaluated the extract of Populus nigra L. in potato plants previously inoculated with P. infestans. They reported a decrease in severity using foliar applications of P. nigra. Islam et al. (2013) observed a yield reduction of 48.2% in tomato plants with the presence of P. infestans and a yield increase of 126% with the foliar application of botanical extracts compared to the control treatment, similar to the results of the present study. The yield increase in tomato plants could be attributed to the compounds present in the A. mexicana extract that have reported biostimulant activity: Hexadecanoic acid, methyl ester; and 9,12,15-Octadecatrienoic acid, methyl ester, (Z,Z,Z; Abubacker & Deepalakshmi, 2013; Kocira et al., 2021). The yield increase seen in this study this may also be due to the foliar application of plant extracts, which has been shown to help plants tolerate stress caused by pathogens, increase product quality, and minimize the need for fertilizers and fungicides (Caruso et al., 2019).

The aerial (Fig. 4E) and dry root (Fig. 4F) weight variables did not show statistical differences between treatments in this study, likely because none of the plants in the study died prematurely, despite some being diseased. All of the plants could continue growing and producing fruits throughout their productive cycle because the Cid F1 variety is indeterminate, which allows apical meristem activity to continue during biotic stress, as one of the plant’s stress response mechanisms. This variety continues growing as part of its survival strategy, trying to produce seeds before the disease causes irreversible damage (Julio et al., 2016; Krishna et al., 2022).

Photosynthetic pigments content in leaves

On average, the content of chlorophyll a, b, and total chlorophyll was reduced by 50% in inoculated plants (Fig. 5). P. infestans causes notable changes at the physiological level, such as a reduction in the photosynthetic rate, changes in transpiration, changes in membrane permeability, an increased respiratory rate, and alterations in tissue expression profiles (Arévalo-Marín et al., 2021). The application of A. mexicana extract increased the content of photosynthetic pigments by an average of 133% (Fig. 5). Naboulsi et al. (2022) reported an increase of 114% in chlorophyll a and 150% in chlorophyll b in tomato plants subjected to abiotic stress and treated with foliar applications of the plant extract based on Crataegus oxyacantha, which aligns with the observations in this study and may be related to the fact that the application of plant extracts contributes to more remarkable synthesis of photosynthetic pigments in leaves (González et al., 2013). Biostimulants of botanical origin, such as DtDREB2A, DtMYB30, DtNAC019, DtNAC72, DtNAC19, DtNAC69, DtZIP63, DtABF3, DtHB12, GRMZM2G439784, GRMZM2G324221, GRMZM2G164 129, and GRMZM2G163866 are involved in signaling events and gene expression and have been reported to transmit signals during plant development, activating defenses against stimuli caused by pathogens, regulating the plant’s response to infections (González-Morales et al., 2021).

Stress biomarkers in tomato leaves

Under optimal growth conditions, levels of reactive oxygen species, including superoxide radicals, hydrogen peroxide (H2O2), hydroxyl radicals (OH•), and singlet oxygen (1O2), are low in plant organelles (Nadarajah, 2020), but increase during stress, damaging proteins and lipids and causing cell damage and death (Verma et al., 2021). In plant-pathogen interactions, high malondialdehyde (MDA) indicate severe oxidative stress (Behiry et al., 2022). Overproduction of reactive oxygen species and lipid peroxidation are related to poor plant growth and fruit quality, and the production of secondary metabolites in commercial crops (Ren et al., 2022). The MDA levels increased by 120% when plants were sprayed with the commercial fungicide Captan, but overall reactive oxygen species and malondialdehyde content were reduced by 31% when plants were sprayed with A. mexicana extract (Fig. 6B). Naboulsi et al. (2022) reported a 44.78% decrease in the MDA content of tomato plants subjected to abiotic stress and hawthorn extract applications (Crataegus oxyacantha L). Behiry et al. (2022) found that MDA and H2O2 levels did not statistically differ with foliar applications of bottle tree extract (Chorisia speciosa St. Hil.) in tomato plants under stress by Rhizoctonia solani compared to the control, which aligns with the findings of this study. The reduced activity could be attributed to the compound n-hexadecanoic acid and dl- α-Tocopherol succinate present in the A.mexicana extract, which has been reported to show activity against reactive oxygen species (Saravanakumar et al., 2018) and improve photosynthetic activity, transpiration rate, stomatal conductance, and antioxidant activity, which are strongly and positively correlated with optimal plant development (Campobenedetto et al., 2021; Hernández-Herrera et al., 2022). Fungicides contribute to membrane lipid peroxidation and mitochondrial dysfunction in plant cells, explaining the increase in MDA seen in plants that received foliar sprays of captan, as chemical fungicides do not discriminate between plant and fungal cells (Gorshkov et al., 2020).

Non-enzymatic antioxidant compounds

The phenol content was reduced by an average of 32% in inoculated plants and increased by 69% with the application of A. mexicana extract (Fig. 7A) because the presence of phytopathogens causes unfavorable changes that compromise the synthesis of non-enzymatic antioxidant compounds, including phenolic compounds (Aina et al., 2022). These findings coincide with the findings of Ertani et al. (2014), who reported that the phenol content increased in pepper leaves (Capsicum chinensis L.) after the application of alfalfa (Medicago sativa L.) and grape (Vitis vinifera L.) extracts. According to the results of this study, as well as those reported in the literature, plant extracts can modify the synthesis of non-enzymatic and enzymatic antioxidant compounds as a result of increased expression of the DET2 gene (Taha et al., 2020), which benefits the defense of the plant in the presence of pathogens (Aitouguinane et al., 2020).

The flavonoid content increased by 113% in tomato plants sprayed with A. mexicana extract (Fig. 7B), a result similar to Giordano et al. (2022), who reported a 141% increase in flavonoid content with the application of tropical plant extracts in lettuce (Lactuca sativa L.) cultivation. The application of plant extracts activates the secondary metabolism by increasing the expression of genes encoding the PAL enzyme, which converts phenylalanine to cinnamic acid and then to coumaric acid. From coumaric acid, flavonoids are synthesized that inhibit pathogen enzymes by chelating the metals necessary for their activity and altering membranes, which affects their function (Schiavon et al., 2010; Ertani et al., 2011; Mierziak, Kostyn & Kulma, 2014). This also helps explain the decrease in disease severity observed in tomato plants inoculated with P. infestans and sprayed with A. mexicana extract.

Vitamin C is a common antioxidant in living organisms, especially plants. It has various functions, such as detoxifying free radicals, regulating plant development, flowering, and responding to pathogens, in addition to acting as an enzymatic cofactor (Egea et al., 2022; Mellidou et al., 2021). The vitamin C content increased at days 45, 60, and 75 after transplanting, when the plant was in the flowering and fruiting stages (Fig. 7C). Although A. mexicana extract application did not generate differences in the vitamin C content at most growth stages, at day 75 ddt, the vitamin C content increased by an average of 133% when the plants were sprayed with A. mexicana compared to the control treatment, coinciding with the findings of Abd El-Hamied & El-Amary (2015) that Moringa oleifera Lam. extract application in pear trees increased vitamin C content.

The variation in vitamin C content observed throughout this investigation is likely because vitamin C content is influenced by the development stage of the plant and by stress (Kukavica & Jovanovic, 2004). In addition, when a phytopathogenic fungus is present, genes such as MDHAR and DHAR are expressed (Paciolla et al., 2019), and specifically in tomato plants, HZ24 is overexpressed. HZ24 is a transcriptional factor that binds to the GMP, GME2, and GGP promoters, which raise vitamin C levels and reduce oxidative stress in plants (Hu et al., 2016).

Carotenoids are natural pigments present in plants that play a crucial role in stabilizing membranes and protecting photosynthesis against damage caused by reactive oxygen species (ROS; Giuliano, 2017; Simkin Andrew, 2021). In this study, β-carotene content increased by an average of 84% in tomato plants that received the foliar application of A. mexicana extract, compared to the Infes treatment (Fig. 7D). These results coincide with Giordano et al. (2022) who reported an increase in β-carotene in lettuce plants (Lactuca sativa) after foliar application of tropical plant extracts. Plant extracts contain phytohormones, carbohydrates, amino acids and proteins that delay the oxidation of β-carotene and stimulate the expression of genes such as GmNAC018, GmNAC030, GmNAC039, and GmNAC043, that delay leaf senescence (Fraga et al., 2021; Yuniati et al., 2022), which is crucial for the biosynthesis of carotenoids and other natural pigments (Azaizeh et al., 2005). Plant extracts offer a promising and sustainable approach for farmers in their agricultural systems (Ali, Ramsubhag & Jayaraman, 2021).

Yellow carotenoids have distinct roles in subcellular organelles, functioning as accessory pigments in chloroplasts photosynthesis and are deposited in different ways in chromoplasts (Khoo et al., 2011; Tanaka, Sasaki & Ohmiya, 2008). The values of yellow carotenoids increased in plants that received the foliar application of A. mexicana extract compared to the other treatments evaluated (Fig. 7E). This coincides with the findings of Khan et al. (2019) that carotenoid content in carrot plants (Daucus carota L.) increased under biotic stress and with the application of Phyllanthus amarus Schumach. & Thonn plant extract. Botanical extracts induce the synthesis of critical enzymes in the mevalonic acid (MVA) and methylerythritol 4-phosphate (MEP) pathways, which are responsible for producing of yellow carotenoids. This biosynthesis is associated with the generation of precursors to plant hormones, such as abscisic acid and strigolactones, together with signaling molecules, such as β-cyclocitral, zaxinone, and ancorene, which influence the development of plants and their ability to resist stress caused by pathogens (Godlewska et al., 2019; Zheng, Giuliano & Al-Babili, 2020; Godlewska et al., 2020).

Proteins are important in plant metabolism and are involved in all the metabolic processes of plants (Sariñana-Aldaco et al., 2022). The protein content did not differ significantly between treatments. Still, it did decrease in tomato plants inoculated with P. infestans and increase in plants that received foliar sprays of A. mexicana extract (Fig. 7F). These results coincide with those of Ertani et al. (2016), who reported that the application of plant extracts from hawthorn, red grape skin, and blueberry increased protein content in corn plants (Zea mays L.) by 115%, on average. When crop stress increases, ROS generation increases, causing more significant oxidative stress, which can decrease the production of proteins (González-Moscoso et al., 2019), explaining the decrease in protein content observed in this study. Conversely, the increase in protein content seen in this study is related to the fact that biostimulants contain plant hormones, such as cytokinins, which reduce the mRNA and protein levels of proteases, preventing an increase in proteolytic activity (Veerasamy, He & Huang, 2007). Promoting the synthesis of cytokinins is beneficial for the crop, but it is vital to use the right type of biostimulant, the correct dose, and optimal application timing (Del Buono, Regni & Proietti, 2023).

Conclusions

This investigation demonstrated that the Argemone mexicana extract contains compounds that have antifungal and biostimulant activity. The presence of Phytophthora infestans reduced tomato crop yield. Still, foliar application of Argemone mexicana extract controlled the severity of the disease by 48% and increased crop yield. A. mexicana extract application also generated an increase in chlorophyll and non-enzymatic antioxidant compounds such as phenols, flavonoids, β-carotene, yellow carotenoids, and proteins, which helped plants tolerate stress caused by P. infestans. These findings suggest that A. mexicana extract may be an ecological alternative to chemical fungicides to mitigate the adverse effects of phytopathogens on tomato crops by inducing growth and activating the antioxidant system.

Supplemental Information

Supplemental Information 1 Data

Click here for additional data file.

The authors would like to acknowledge the Universidad Autónoma del Estado de Hidalgo (UAEH) and the Universidad Autónoma Agraria Antonio Narro (UAAAN) for allowing us to conduct the study in their facilities. This study formed part of Iridiam Hernández-Soto’s PhD thesis (Doctorado en Ciencias Agropecuarias, Universidad Autónoma del Estado de Hidalgo).

Additional Information and Declarations

Competing Interests

Author Contributions

Data Availability

The authors declare there are no competing interests.

Iridiam Hernández-Soto performed the experiments, authored or reviewed drafts of the article, and approved the final draft.

Yolanda González-García analyzed the data, prepared figures and/or tables, authored or reviewed drafts of the article, and approved the final draft.

Antonio Juárez-Maldonado analyzed the data, authored or reviewed drafts of the article, and approved the final draft.

Alma Delia Hernández-Fuentes conceived and designed the experiments, authored or reviewed drafts of the article, and approved the final draft.

The following information was supplied regarding data availability:

The raw measurements are available in the Supplemental File.

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
