# Peer review of "Impact of Argemone mexicana L. on tomato plants infected with Phytophthora infestans"

_PeerJ, doi:10.7717/peerj.16666_

## Round 0.1 · original submission · Major Revisions

Dear Dr. Hernández Soto:

Thank you for submitting your manuscript to PeerJ for publication. Two great experts in the Phytophthora research community have reviewed your manuscript and their excellent comments are attached for your reference. In particular, reviewer 1 raised some concerns about your experimental designs, treatments, and comparative data analysis. More importantly, results related to unaffected plants. In addition, Fig. 1A and 1B need to be addressed clearly. Also, please describe your Discussion section more precisely and concisely how your results are similar or different from the previous works.

Best regards,

Sincerely,

Tika Adhikari

**Language Note:** PeerJ staff have identified that the English language needs to be improved. When you prepare your next revision, please either (i) have a colleague who is proficient in English and familiar with the subject matter review your manuscript, or (ii) contact a professional editing service to review your manuscript. PeerJ can provide language editing services - you can contact us at [email protected] for pricing (be sure to provide your manuscript number and title). – PeerJ Staff

Reviewer 1 ·

Basic reporting

This manuscript reports the effect of spraying an extract of a weed native to Mexico, Argemone mexicana, on tomato. Results include the effect of the compound on resistance to the oomycete pathogen Phytophthora infestans, and on vegetative growth, fruit yield, and the accumulation of chlorophyll, potential antioxidants, and compounds related to oxidative stress. The weed extract is reported to reduce disease incidence, modestly increase growth and fruit yield, and result in higher levels of the antioxidants and in some cases lower levels of the markers of oxidative stress such as MDA.

Some of the results are interesting, although not entirely novel as prior studies (using weed or algal extracts) have reported similar results. Our ability to interpret the results are also limited since the content of the extracts is not described. For example, prior studies have shown that simply spraying amino acids on plants can yield similar growth-promoting effects. Whether the Argemone extracts have effects due to other types of compounds can thus not be determined.

Overall, I found that most of the experiments were performed suitably and had few concerns about the results involving uninfected plants.

Experimental design

Overall, I found that most of the experiments were performed suitably and had few concerns about the results involving uninfected plants. However, I found the results using infected plants (with P. infestans) difficult to fathom, and much clarification is needed. This may affect the validity of the findings, as described next.

Validity of the findings

For example, Fig. 1A shows nicely how the extract reduced the disease phenotype. But I am confused by how plants showing a disease score of 90% long before the date of fruit set would show no difference in traits such as fruit yield or leaf weight over the course of the experiment (the disease score essentially means the amount of leaf surface area showing necrosis). This needs to be addressed. Since the El Cid cultivar of tomato is indeterminate, did growth of the surviving parts of the plant accelerate to catch up with the control plants? Did any plants die early during the experiment that were removed prior to the later timepoints?

Also, Fig. 1B shows no obvious P. infestans lesions in any of the plants treated with Captan or the weed extract. I found this surprising since the plants were infected 15 days before the extract (and presumably captan) was added. As stated early in the paper (li 50) P. infestans would cause total necrosis of the plant within 5-10 days. Since the extracts were added at day +45 after transplanting, with the pathogen added at day +30, why can no symptoms of infection can be seen in the images? What were the conditions in the greenhouse? Is it possible that the temperature was too high for the pathogen to grow? Are there small lesions on the plants that are not visible in the picture? Were the conditions (temperature and humidity) conducive to sporulation in the greenhouse, which would initiate multiple cycles of infection? When were these pictures taken? What fraction of the leaves were actually infected? Related to the latter, it is possible that the infection rate was low since the cultures were incubated at a temperature (27C) that is high for most strains of P. infestans, reducing sporangia viability; although it is possible that their isolate was more thermotolerant than most members of the species.

Besides providing more information about the incubation conditions, which might help the reader interpret the results, other experimental details need to be added to the manuscript. These include:
--the captan dosage;
--the strain of P. infestans;
--the disease rating scale; Materials and Methods says that they used a method modified from a system used with another tomato pathogen in the Zarate paper, but the nature of the modifications was not described.
--how the weed extract was resolubilized.
--the number of plants per block and treatment
--when the Fig. 2 data were collected (presumably at +105 days

Additional comments

Comments about the writing style:
Other than some of the wording issues such as those mentioned below, in general the manuscript was structured reasonably well. An exception was Discussion. This was overly long, almost like a series of minireviews addressing many of points related to the paper. For example, most of the first paragraph of Discussion described the role of molecular factors of the pathogen (such as effectors) in causing disease. None of this seems particularly relevant to the paper. Later, whole paragraphs are used to summarize potential ways in which a weed extract might affect a plant, genes affected by plant extracts, stress biomarkers, effects of Vitamin C, etc. I am not saying that these should be entirely deleted but many of these paragraphs could each be condensed into 1-2 sentences.

Minor points:
li 44. Annually?
li 45. Phytophthora is an oomycete, not a fungus.
li 61. Are extracts actually used by growers, or just described by researchers? Probably better to say "may yield."
li 64. Capitalize C in commonly.
multiple locations: use lower case after semicolon. Or replace the semicolon with a period and keep the uppercase (. T, not ; T).
li 119. Use sporangia, not conidia.
li 168. Service?
li 191. measure not Measure
li 238. clarify if yield per plant or weight per fruit is being referred to.

Reviewer 2 ·

Basic reporting

The manuscript is well-structured and professionally written, containing sufficient and engaging literature for readers. Its focus on an economically significant plant is likely to attract readers' attention.

Experimental design

Aim of the research well achieved but I have few questions on experimental design that mention below
Materials and Methods:
Study area map should be added in the plant material:
Obtaining the plant extract:
How author ensure that (1:10) ratio is enough for the treatment of pathogen?
Crop development and management:
Could you describe the irrigation system employed for the tomato plants in terms of different growth stages?
Application of treatments
What are the different treatments applied to the tomato plants in the study, and how was the chicalote extract administered? Can you also explain the timing and frequency of the applications?

Validity of the findings

Results:
P. infestans severity and crop development:
Which treatments remained disease-free throughout the evaluation period, and why was this outcome expected?
What were the changes in the number of fruits per plant in response to the Infes treatment and the treatments involving EXAm and captan?
Non-enzymatic antioxidant compounds:
Among the treatments, which one exhibited the highest increase in phenol content during the flowering and fruit set stages? What was the percentage increase compared to the control treatment?

Additional comments

The discussion is too long; it should be concise.
Conclusion should be re-written.

---

## Round 0.2 · Minor Revisions

Dear Dr. Hernández Soto,

I am writing to inform you that your manuscript - Impact of Argemone mexicana L. on tomato plants infected with Phytophthora infestans - still needs some minor corrections. Please see below comments from our section editor.

1) "There were nine compounds identified in this extract with antifungal and biostimulant activity": Maybe these compounds are suggested to have those activities. Have they been tested against P. infestans; or as biostimulants? Which other compounds were found in the GC-MS analyses?

2) Many redundant phrases can be eliminated for improving the readability, e.g.: "In the present study," "The results of this study showed that the application, .." just: "The application...""

I would greatly appreciate it if you could use the attcahed file to revise your manuscript and resubmit it at your the earliest possible time.

Thank you for your understanding.

Best regards,

Tika Adhikari

Reviewer 2 ·

Basic reporting

I have checked thoroughly the revised manuscript. Now it's in the form of consideration for publication.

Experimental design

All technical changes in experiments are done by author in revised manuscript so it could be published.

Validity of the findings

It's novel research and will provide benefits to human society.

---

## Round 0.3 · accepted · Accept

Dear Dr. Hernández Soto,

Thank you for your submission to PeerJ.

I am writing to inform you that your manuscript - Impact of Argemone mexicana L. on tomato plants infected with Phytophthora infestans - has been Accepted for publication.

Congratulations!